# Improving Printability of Digital-Light-Processing 3D Bioprinting via Photoabsorber Pigment Adjustment

**DOI:** 10.3390/ijms23105428

**Published:** 2022-05-12

**Authors:** Jeong Wook Seo, Gyu Min Kim, Yejin Choi, Jae Min Cha, Hojae Bae

**Affiliations:** 1Department of Stem Cell and Regenerative Biotechnology, KU Convergence Science and Technology Institute, Konkuk University, Seoul 05029, Korea; wjddnr9302@naver.com (J.W.S.); ggyu1005@gmail.com (G.M.K.); blessyejin16@gmail.com (Y.C.); 2Department of Mechatronics Engineering, College of Engineering, Incheon National University, Incheon 22012, Korea; j.cha@inu.ac.kr

**Keywords:** photorheology, DLP, bioink, bioprinting, photoabsorber

## Abstract

Digital-light-processing (DLP) three-dimensional (3D) bioprinting, which has a rapid printing speed and high precision, requires optimized biomaterial ink to ensure photocrosslinking for successful printing. However, optimization studies on DLP bioprinting have yet to sufficiently explore the measurement of light exposure energy and biomaterial ink absorbance controls to improve the printability. In this study, we synchronized the light wavelength of the projection base printer with the absorption wavelength of the biomaterial ink. In this paper, we provide a stepwise explanation of the challenges associated with unsynchronized absorption wavelengths and provide appropriate examples. In addition to biomaterial ink wavelength synchronization, we introduce photorheological measurements, which can provide optimized light exposure conditions. The photorheological measurements provide precise numerical data on light exposure time and, therefore, are an effective alternative to the expendable and inaccurate conventional measurement methods for light exposure energy. Using both photorheological measurements and bioink wavelength synchronization, we identified essential printability optimization conditions for DLP bioprinting that can be applied to various fields of biological sciences.

## 1. Introduction

Recently, three-dimensional (3D) bioprinting technology, using a digital light processing (DLP) 3D printer, has attracted increased attention owing to its ability to generate replicas of complex organ structures, including artificial organ models [1], vascular network formation [2], and microscopic liver tissue [3], while also demonstrating success in various biomedical applications [4,5,6]. Projection base bioprinters reproduce 3D geometrical structures through light exposure controlled by the digital processing of photocrosslinkable biomaterial ink. They are suitable for fabricating large-scale scaffolds with complex structures, as they use reverse-gravity stacking, precise photocrosslinking technology, and rapid printing processes [7,8,9]. Compared with other 3D printers, such as extrusion-based 3D printers, a highly photoreactive biomaterial ink is required for projection base 3D printers, as they use techniques based on the principle that photocrosslinking properties are controlled by light exposure [10]. Therefore, an emerging need exists for an optimized biomaterial ink method to ensure successful bioprinting.

The existing hydrophobic resins for general DLP printing have excellent printability and high printing accuracy; however, the resin components are unsuitable for bioprinting [1]. Compared to the current commercially available hydrophobic resin, the biopolymer-based bioink used for bioprinting is superior in terms of low toxicity, biocompatibility, and biomimetic properties; however, it lacks printing accuracy. As a cell-friendly environment needs to be maintained, the use of various hydrophobic photoinitiators is limited in bioinks that must remain in a hydrophilic solution phase [11]. Nevertheless, suitable photoinitiators have been identified for bioink. For instance, lithium phenyl-2,4,6-trimethylbenzoylphosphinate (LAP), a photoinitiator widely used in DLP bioprinting, is known for its hydrophilic properties and low cytotoxicity [12,13,14]. LAP also has relatively rapid polymerization kinetics, making it suitable for cell-encapsulation applications [15]. When a photoinitiator is exposed to the light energy of a reactive wavelength, it generates free radicals that initiate photopolymerization with the polymer in the bioink [16,17]. However, LAP is suited for the absorption of UV and blue-light wavelengths (350–400 nm); the UV absorption range has been known to negatively affect cell behavior and generate excessive heat [18,19]. In addition, the relatively wide absorption wavelengths of bioinks are often incompatible with the wavelength band of the commercially available printer projection light; this may affect printability. Therefore, it is necessary to control the absorption wavelength region to overcome these challenges when using LAP.

The optimization of light exposure energy is a key requirement for the DLP bioprinting process. In this study, a 3D structure of hydrogel was printed as Z-stacks, with each focal layer set to a thickness of 100 µm. In this process, a low level of light exposure energy cannot pass beyond 100 μm, thus preventing the formation of the hydrogel. Conversely, excessive light exposure energy causes light to penetrate beyond the focal layer, resulting in over-crosslinking, which leads to an undesired shape. Recently, a multi-hollow channel test was employed to assess over-crosslinking in DLP bioprinting [19]. However, to calculate the optimal amount of exposure energy in this method, a trial-and-error process, comprising the modification of light exposure energy for different condition values, is essential. This leads to wasted biomaterial ink components. Hence, a method is required that can assess the optimization conditions of biomaterial ink before printing. To this end, we propose the use of photorheological measurements as an alternative method to obtain precise numerical data.

The purpose of this study was to synchronize the light wavelength of the commercially available printer and the absorption wavelength of the bioink, while optimizing the light exposure energy to maximize printability and cell viability. Specifically, to address the challenge associated with incompatible 3D printer and bioink wavelengths when using LAP as a photoinitiator, we investigated the light wavelength of the projection light and the absorption wavelength of the bioink with various colors of photoabsorbers (red, orange, yellow, green, blue, purple, pink, and black). The photoabsorber color was selected by comparing the projection light wavelength and absorption wavelength. Next, the process of over-crosslinking was systematically analyzed, and conventional optimization methods of light exposure energy were improved by performing photorheological measurements. The gelation time was then measured by using a selected yellow photoabsorber, and a hollow space model was printed to validate the selected photoabsorber and designed photorheological measurements. Finally, the controllability of the light transmittance was confirmed through a mechanical compression test.

## 2. Results and Discussion

### 2.1. Color of Photoabsorber

To optimize the photoabsorber color for the DLP bioprinting process, a photocrosslinkable biomaterial ink comprising poly(ethylene glycol) dimethacrylate PEGDMA and LAP was mixed (Figure 1A). For bioprinting, it is necessary to synchronize the peak absorption wavelength of the photoinitiator and the wavelength band of the projection light of the printer. In this study, the wavelength band of the UV-Vis source of the DLP projection light was between 380 and 450 nm, with a peak occurring at 405 nm (Figure 1B). However, the wavelength of the biomaterial ink absorption peak produced without a photoabsorber was 300–350 nm (Figure 1C).

Biomaterial inks with unsynchronized absorption wavelengths transmit or reflect projection light energy, resulting in over-crosslinking, which causes photocrosslinking in unwanted areas. To circumvent this issue, a photoabsorber that inhibits light transmission and reflection, while allowing energy to be absorbed by the biomaterial ink, is required [20]. The absorption wavelength was determined according to the pigment color, using a spectrophotometer. The color found to be the most synchronized with 405 nm, the main peak wavelength of the 3D printer projection light, was yellow [21] (Figure 1D). Subsequently, the same model was printed according to the color of the light absorber. The difference in absorbance, according to color, was noticeable, and the printing result of the yellow sample most resembled the template (cylinder). By contrast, the pink and blank groups experienced severe over-crosslinking, resulting in poor printing (Figure 1E).

The printability ratio (%) is a number that represents the difference between modeling and actual printing; the closer the printability ratio is to 100%, the more accurately the programmed modeling was implemented (Figure 1F). The printing of the yellow sample was closest to the modeling, and the printability ratio of the yellow group was 96.49 ± 2.89%. Next, the printability ratio of the orange and blue samples were 102.41 ± 7.62% and 115.79 ± 5.32%, respectively. There was no statistically significant difference between the yellow, orange, and blue groups. The pink and blank groups produced printing results that were significantly different from the input modeling, with printability ratios of 145.18 ± 24.1 and 153.83 ± 26.86%, respectively. There was a statistically significant difference between the yellow group and the pink (**) and blank (***) groups. The results confirmed that the differences in the printability ratio of the colors were due to there being more areas synchronized in the 3D printer’s light wavelength band (Figure 1C) and absorbance according to color (Figure 1D). The samples showed printing results according to synchronization between the wavelength band of projection light and the absorption wavelength band of each color. Among these samples, the yellow group showed the highest synchronization and a high level of modeling fidelity, whereas the pink group, with the lowest synchronization, exhibited low absorbance.

### 2.2. Effect of Photoabsorber on Over-Crosslinking

Biomaterial ink that is not synchronized with the projected light of a 3D printer cannot absorb the irradiated light, causing unavoidable refraction. The refracted light spreads out, causing unwanted photocrosslinking (Figure 2A,B). We printed an unsynchronized biomaterial ink (pink) and a synchronized biomaterial ink (yellow) in the same 3D square-shaped hollow tube design (Figure 2C, 1000 μm × 1000 μm and 900 μm × 900 μm, respectively). The process of printing a hollow model can be explained in four steps (Figure 2A, control vs. yellow).

First, to initiate printing, the polymerization plate is set at a height of one layer (100 μm) in a bioink tank. The biomaterial ink is then photocrosslinked by the DLP light, which originates from the bottom of the printer, according to the programmed shape (pink arrows). At this step, if the wavelength band of the DLP light and the absorption band of the biomaterial ink are not synchronized, the DLP light refracts and penetrates at an undesirable angle (blue arrows). Correspondingly, exposure to light in an undesired direction leads to over-crosslinking (Figure 2A, control, step 1).

Second, over-crosslinking within the hollow tube area takes place, as the unsynchronized biomaterial ink does not absorb the light in an efficient manner, due to irregular DLP light exposure, thereby blocking the hollow space (Figure 2A, control, step 2).

Third, the light that is not absorbed by the biomaterial ink continuously affects the previously crosslinked hydrogel layer (Figure 2A, control, step 3), thereby blocking the hollow tube space (Figure 2A, control, step 4). On the contrary, in the case of synchronized biomaterial ink (yellow), the hollow tube model printed the corresponding design (Figure 2A, yellow, step 4).

In the resultant print of the control biomaterial ink (pink), the originally designed 3D modeling was ignored owing to the printing of components that were not part of the desired shape (Figure 2B). In fact, the hollow tube hydrogel printed with pink biomaterial ink was completely blocked due to over-crosslinking, which was not observed with the use of the yellow biomaterial ink (Figure 2C). 

### 2.3. Photorheological Measurement

Photorheological measurements were obtained by using a rheometer to realize optimal printability by measuring the light exposure energy optimized for the biomaterial ink. Photorheological measurements can analyze the behavior of the sol-gel transition that occurs as the light energy gradually increases in the biomaterial ink. The storage modulus (G′) and loss modulus (G″) were measured as a function of light exposure time (Figure 3A). The biomaterial ink, loaded at a height of one layer (100 μm) between the rheometer measuring glass and the plate, exhibits the typical rheological behavior of a liquid (sol) phase (Figure 3A, (1)). During this phase, G″ was greater than G′, indicating that the polymer (GelMA) was completely dissolved in the solution. As the photocrosslinking begins, the liquid biomaterial ink starts to enter the sol-gel transition state (Figure 3A, (2)). The sudden decrease in G′ seen in this phase is likely due to the formation of the heterogeneous polymer, making it difficult for the rheometer to provide accurate measurements [22]. As the biomaterial ink reaches the gelation point (~12 s), both G′ and G″ increased with G′ > G″, indicating the formation of a viscoelastic hydrogel (Figure 3A, (3)).

As the gelation time represents an optimized exposure-light intensity that can minimize over-crosslinking, we investigated the gelation time of different photoabsorber concentrations by conducting photorheological measurements. Our investigation confirmed that more energy was required to induce gelling at a depth of 100 μm, as the light absorption increased with an increase in the photoabsorber concentration. The blank (0%) group without a photoabsorber exhibited an excessively low gelation time of 6.83 ± 0.02 s. Conversely, the yellow (3%) photoabsorber group had a high gelation time of 17.47 ± 1.51 s (Figure 3B). Accurate printing was achieved at the gelation time for all concentrations, with the exception of the 0% and 0.5% groups, for which the absorption wavelengths were not synchronized, due to low absorbance. The gelation time showed a dose-dependent delay in the induction of photocrosslinking.

Moreover, the use of photoabsorbers at high concentrations has the potential to cause deviations in high-precision printing results, as was confirmed when the printing process was carried out without the application of an optimized gelation time. To do this, three different light exposure times (9, 11, and 13 s) were applied for the printing of a 3D square-shaped hollow tube design (Figure 3C). The sample that induced under-crosslinking did not achieve complete photocrosslinking, and a hollow space larger than that of the model was formed (Figure 3C, left). This formation was structurally unstable, as it did not achieve complete gelling. In contrast, in the sample that induced over-crosslinking, hydrogel formed in unwanted areas due to excessive photocrosslinking (Figure 3C, right). Excessive hydrogel formation was observed as a thin layer of film in the hollow space region. Finally, the optimized crosslinking sample produced by using the optimized gelation-time data exhibited perfect printability for modeling, structural stability, and distinct formation of the hollow spaces (Figure 3C, center).

### 2.4. Mechanical Properties

The mechanical properties of a specific hydrogel are extremely important in selecting its application. Therefore, we evaluated the hydrogels according to the different photoabsorber concentrations (0.5, 1, 1.5, 2, 2.5, and 3%) under the same cumulative light exposure energy (34.42 mW/cm^2^). The 0.5% group yielded the highest modulus value of 400 ± 34.93 kPa, as the low photoabsorber concentration enables light to cross the focal layer, which continuously induces photocrosslinking of the previously printed hydrogel layer during the printing process. In contrast, the 3% group yielded a modulus of 104.27 ± 31.88 kPa; as the concentration of the photoabsorber increased, the absorption increased, and the photocrosslinking range was gradually limited to the focal layer, resulting in a decreased compressive modulus. The samples without any photoabsorber (0% group) were not successfully processed, as they did not meet the required measurement conditions. Collectively, these results confirm that the photoabsorber and compressive modulus have an inverse relationship under the same light exposure energy (Figure 4A).

An inverse decreasing trend was also observed in the stress–strain curve. That is, in the initial strain (~30%), a clear difference was observed between the groups. The 0.5% group, which had the lowest concentration of photoabsorber, exhibited the fastest increasing trend, whereas the 3% group showed the slowest (Figure 4B). Therefore, these results indirectly confirm that the refraction and penetration of light can be controlled by the concentration of the photoabsorber through the compressive modulus.

### 2.5. Cell Viability

The cell viability of encapsulated bovine ear fibroblast cells (BEFCs) was investigated for one day to determine the effect of the photoabsorber color (pink vs. yellow) and optimized light exposure time (11.8, 21.8, and 31.8 s) on cell viability. The cells were first exposed to light for 11.8 s, as this is the gelation time of the bioink with the yellow photoabsorber, and then exposed to two additional longer light exposure times (21.8 and 31.8 s) to investigate the change in cell survival rate under harsh conditions. At 11.8 s exposure time, the yellow group did not exhibit significant cytotoxicity against cells on the hydrogel (88.05 ± 2.11%). In addition, many cells exhibited adhesion and were successfully cultured. In contrast, following light exposure for 11.8 s, the pink group exhibited 45.69 ± 17.92% cell viability, and cell adhesion was not observed. In the case of 21.8 s exposure, the yellow group was in a stable live state (81.6 ± 2.5%); however, few cells exhibited adhesion. Meanwhile, in the 21.8 s light-exposure pink group, most cells appeared to be dead (7.76 ± 5.06%) [23,24,25]. Most cells in the 31.8 s light-exposure yellow group were also in a stable live state; however, the relative number of dead cells had increased (75.36 ± 1.15%). The 31.8 s exposure pink group comprised primarily dead cells (1.21 ± 0.92%), due to the excessive light exposure (Figure 5A,B). 

Through using the pink photoabsorber groups as controls, we confirmed that the absorption wavelength synchronization affects not only printability but also cell viability. In addition, we confirmed that the excessive exposure of bioink to light with synchronized absorbance can negatively affect cell viability and adhesion.

### 2.6. The Printed Structures Fabricated with Optimized Printing

Complex structures were printed to confirm the applicability of the optimized printing strategy. The yellow photoabsorber was most synchronized with the wavelength of the printer, and the light exposure time optimized by photorheological analysis made it possible to print complex structures. The optimized printing environment, which enabled detailed description of the curved or hollow structure of an organ, was able to express a complex and curved structure similar to that of a human ear (Figure 6A). However, in the case of the pinna, it was possible to design and print the supporter structure. It was also possible to print microchannels requiring high resolution. The round-type microchannel could not print less than 250 µm, but it was successfully expressed from 300 to 1000 μm (Figure 6B). Similarly, the square-type microchannels were printed in a blocked state under 200 µm. The planned modeling without blockage in the flow of the fluid from 300 um to 1000 µm was well expressed (Figure 6C).

## 3. Materials and Methods

### 3.1. Materials

PEGDMA (MW: 1000) was purchased from Polysciences (Warrington, PA, USA). Edible pigments were purchased from Lgreentech (Daejeon, Gyeonggi, South Korea). Penicillin/streptomycin (P/S) and phosphate buffered saline (PBS, pH 7.4) were purchased from WelGene (Daegu, Gyeongbuk, South Korea). Gelatin (Type A, 300 bloom from porcine skin), methacrylic anhydride (MA), and LAP (≥95%) were purchased from Sigma-Aldrich (St. Louis, MO, USA). All other chemical agents used in this study were of analytical grade.

### 3.2. Biomaterial Ink Preparation and DLP Printing Process

To prepare the biomaterial ink, PEGDMA was completely dissolved (10% [*w*/*v*]) in PBS with 1% P/S and 0.5% (*w*/*v*) LAP. A photoabsorber was added according to the experimental conditions. The prepared biomaterial ink was then transferred to a bioink tank of IM2 (DLP 3D printer; Carima, Seoul, Korea). All printing was performed by setting the light intensity of the printer to 1.97 mW/cm^2^. The thickness of a single layer was set to 100 μm for printing. The initial three layers were exposed to light for a longer period (+1 s) to prevent the photocrosslinked 3D constructs from becoming separated from the plate (Figure 1A). All printing models were designed by using Fusion 360 software (Autodesk, San Rafael, CA). An 8 mm × 2.5 mm (diameter × height) model cylinder was printed for each sample (yellow, orange, blue, pink, and blank). A printability rate was calculated, comparing the volume of the printed sample with the volume of the model:
Printability rate = volume of printed samples/volume of modeling × 100 (%).

### 3.3. UV-Vis Spectrometer Measurement

Absorbance measurement was performed by using an Agilent 8453 UV-Vis (Agilent technologies, Santa Clara, CA, USA). After mixing 1% (*v*/*v*) photoabsorbers of various colors (Red, Orange, Yellow, Green, Blue, Purple, and Pink) in DPBS with 0.5% (*w*/*v*) LAP, the samples were placed in a cuvette and measured. The blank group did not contain any photoabsorber.

### 3.4. Photorheological Measurement

Photorheology was performed by using a HAKKE MARS 40 (Thermo Fisher Scientific, Waltham, MA, USA) equipped with a UV module accessory. To investigate the rheological behavior of the biomaterial ink when the DLP printer was irradiated with light, a light source with a wavelength similar to that of IM2 was produced by an OmniCure LX 505 (Lumen Dynamics; Mississauga, ON, Canada), equipped with a 405 nm light-emitting diode (LED) channel. The light intensity was 2.45 mW/cm^2^. Radiation was guided through the collimator and reflected toward the measuring glass (Figure 3A). The LED intensity was measured by using a rheometer measuring glass and set to the same intensity as that of IM2. Biomaterial ink was loaded at 100 μm intervals between the glass and the bottom plate. The oscillation experiment was conducted with an oscillating shear strain of 0.08% at a frequency of 20 Hz.

### 3.5. Mechanical Properties

A CT3 Texture Analyzer (Brookfield; Toronto, ON, Canada) with a 4500 g load cell (Brookfield; Toronto, ON, Canada) in compression mode was used to measure the compressive strength of the hydrogels. All measurement samples were printed with a light exposure time of 17.47 s, which is the gelation time point for the 3% group. A 12.7 mm-diameter probe was used for compression with a trigger load of 0.05 N and a test speed of 0.05 mm/s. The compressive modulus was determined as the slope of the linear region corresponding to a 5–15% strain [26]. All measurements were performed by using a DLP-printed cylinder model (8 mm (d) × 2 mm (h)). 

### 3.6. Cell Viability

For DLP bioprinting, Gelatin methacrylation (GelMA) was synthesized as described previously [1]. Regarding the preparation of bioink, lyophilized GelMA was completely dissolved (10% [*w*/*v*]) in DPBS at 37 °C with 2% (*v*/*v*) FBS, 1% penicillin/streptomycin (P/S), 0.5% (*w*/*v*) LAP, and 1% (*w*/*v*) yellow or pink pigment. Cultured BEFCs were detached from the culture plate and added to the GelMA solution at a concentration of 1.5 × 10^6^ cells/mL. The prepared bioink was then transferred to an IM2 (DLP 3D printer; Carima, Seoul, Korea) bioink tank that was preheated to 37 °C. The thickness of a single layer was set to 100 µm. Each sample was printed with three light exposure times (11.8, 21.8, and 31.8 s). All measurements were performed by using a DLP-printed cylinder model (8 mm (d) × 2 mm (h)). After the printing process, the scaffold was removed from the plate and washed twice with DPBS (37 °C). Finally, the scaffold was cultured in high-glucose DMEM supplemented with 10% FBS and 1% P/S for 1 day. Cell viability was investigated by using calcein AM/ethidium homodimer live/dead assay kits (Invitrogen, Carlsbad, CA, USA). After 1 day of incubation, 1 mL of the staining solution was added according to the manufacturer’s protocol, and cells were imaged by using a Lionheart FX microscope (BioTek Instruments, Winooski, VT, USA). The imaged cells were counted by using Gen5 software (supplied with the Lionheart FX; BioTek Instruments, Winooski, VT, USA). The ratio of live cells to the total number of cells was used as a metric of cell viability.

### 3.7. Statistical Analysis

The results were presented as means ± standard deviations (SDs). All statistical analyses were carried out by using the GraphPad Prism 8.0.2 program (GraphPad Software; La Jolla, CA, USA). One-way analysis of variance (ANOVA) with Tukey’s post hoc test was used. Iteration numbers of each experiment can be found in the corresponding figure captions; * *p* < 0.05, ** *p* < 0.01, and *** *p* < 0.001 were used to indicate the significance.

## 4. Conclusions

In this study, the relationship between the color and absorption of a photoabsorber was investigated for eight edible pigment candidates, using the commercially available projection base printer. The peak wavelength band of the DLP projection light and the absorption wavelength of the biomaterial ink were investigated. It was possible to control the absorption peak wavelength through the color of the photoabsorber. An optometric analysis for DLP printing was devised through a stepwise breakdown of the over-crosslinking phenomenon. Photorheological measurements showed the possibility of providing optimized printing light exposure conditions with precise measurement results. A rather low printability of bioink with excellent biocompatibility was controllable via the absorption wavelength band and optimized light exposure conditions. Moreover, we found that successful DLP bioprinting with cells requires not only synchronization of the absorption wavelength but also determination of the minimum light exposure time, that is, the gelation time. This bioink optimization method can increase the usability of various biomaterials and is expected to be utilized in fields such as microfluids [27] and vascularized tissue engineering [28,29], which require high printability.

## Figures and Tables

**Figure 1 ijms-23-05428-f001:**
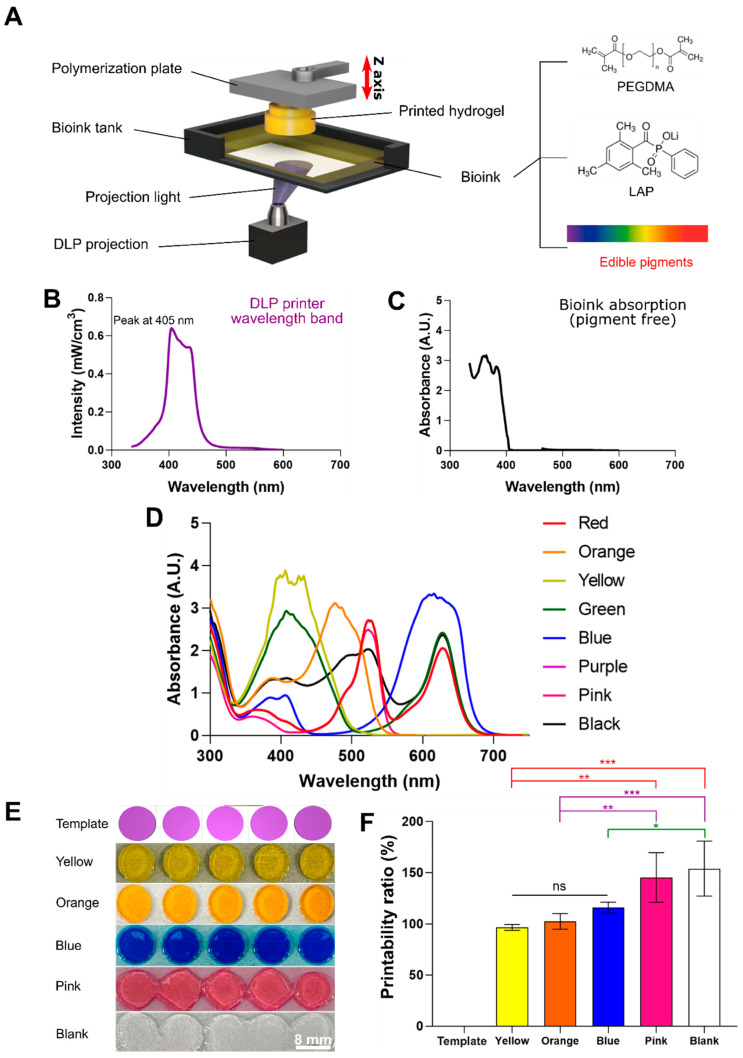
(**A**) Schematic diagram of DLP bioprinting and absorption analysis according to the DLP projection light wavelength and photoabsorber color. The projection light illuminates the bioink tank from below, causing biomaterial ink to build up (units of 100 μm) under the polymerization plate while inducing photocrosslinking. The biomaterial ink consists of a polymer (poly (ethylene glycol) dimethacrylate (PEGDMA)), photoinitiator (LAP), and photoabsorber (edible pigments). (**B**) Intensity of the projection light at different wavelengths of the UV-Vis spectrum. The peak is observed at 380–450 nm. (**C**) Absorbance of biomaterial ink without edible pigments, which are photoabsorbers. (**D**) Absorbance of biomaterial inks with photoabsorbers of different colors. The wavelength range of the absorption peak differs for each color. (**E**) Image depicting the designed template (cylinder shape) and printing according to the photoabsorber color (yellow, orange, blue, pink, and blank). (**F**) Comparison of printability ratio of light absorbers of different colors. Data are presented as the means ± standard deviations (*n* = 5; * *p* < 0.05, ** *p* < 0.01, and *** *p* < 0.001; ns, not significant) and analyzed by a one-way ANOVA followed by Tukey’s post hoc test.

**Figure 2 ijms-23-05428-f002:**
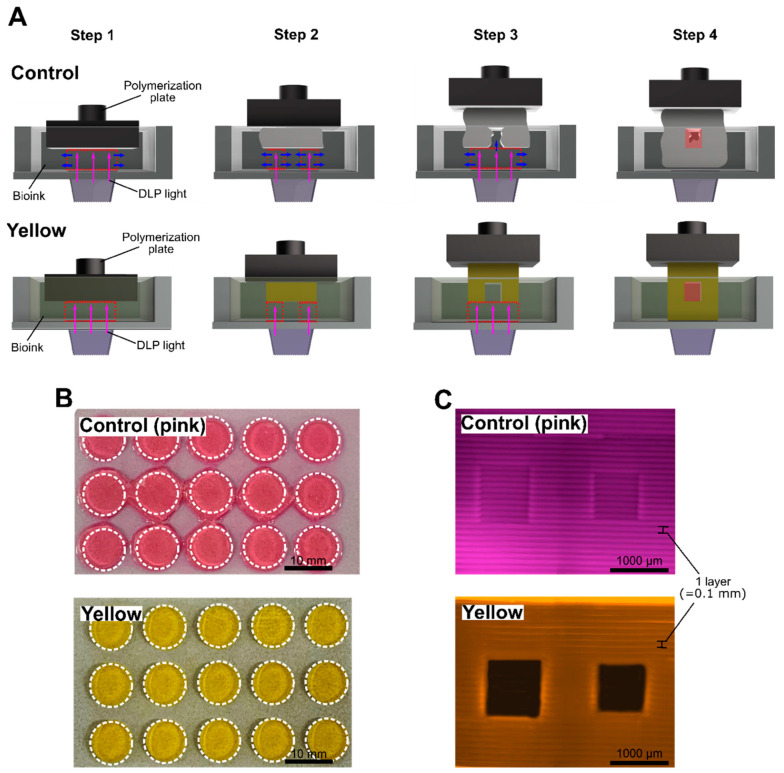
(**A**) Schematic diagram depicting the process of printing a hydrogel with a hollow space, using biomaterial ink with an unsynchronized absorption wavelength with DLP light (control) and with a synchronized absorption wavelength (yellow). The red dotted blank space indicates the focal layer, the purple arrow indicates the direction of the DLP light, and the blue arrow indicates the light due to the refraction and penetration of the DLP light. (**B**) Sample prints resulting from pink and yellow biomaterial ink (top view). In the control samples, hydrogels were printed to unwanted areas due to over-crosslinking. (**C**) Print samples resulting from pink and yellow biomaterial inks (front view). In the pink samples, hydrogels blocked hollow tube areas due to over-crosslinking.

**Figure 3 ijms-23-05428-f003:**
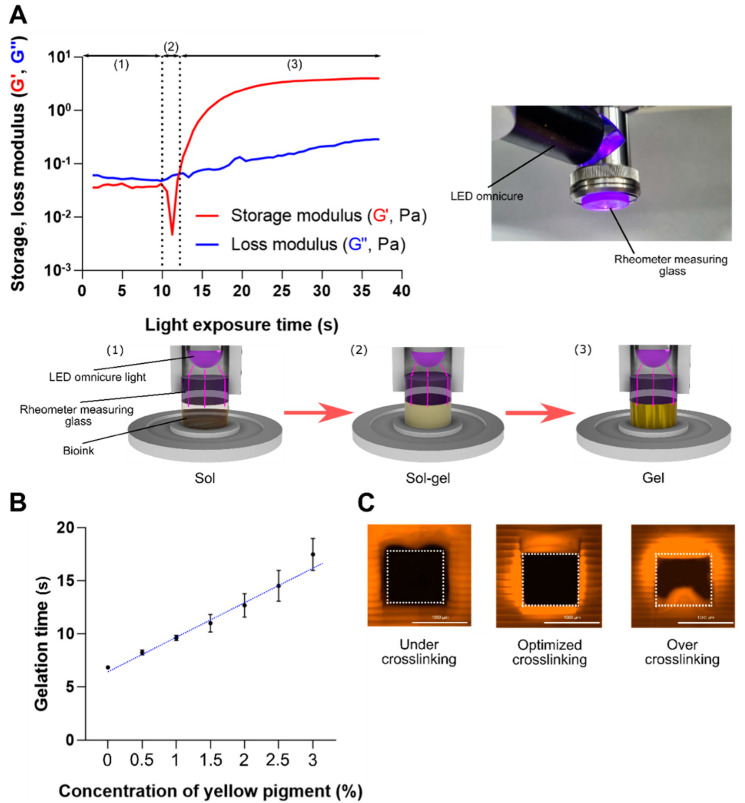
(**A**) Representative photorheological analysis graph of sol-gel phase change (storage modulus (G′) and loss modulus (G″)) and schematic diagram describing the photorheological measurement process. Region (1), biomaterial ink is in liquid (sol) phase; region (2), some biomaterial ink starts to form aggregate; region (3), gel phase. (**B**) Gelation time graph of the yellow photoabsorber concentrations (0, 0.5, 1, 1.5, 2, 2.5, and 3%). Data are shown as the mean ± SD, *n* = 3. (**C**) Results of under-crosslinked (9 s), optimized (11 s), and over-crosslinked (13 s) printed samples. Scale bar = 1000 μm.

**Figure 4 ijms-23-05428-f004:**
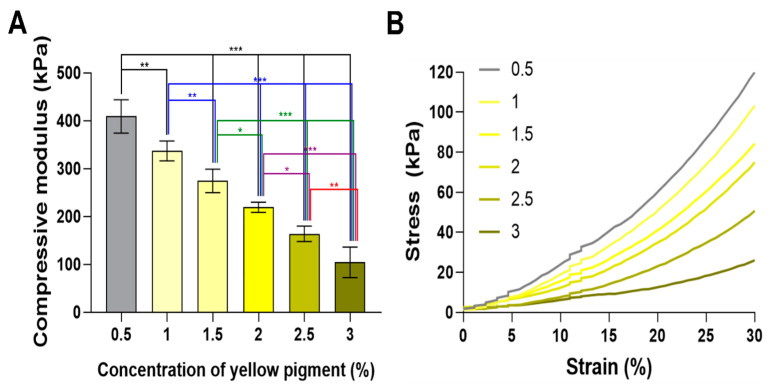
(**A**) Mechanical properties of printed hydrogels with various concentrations of yellow photoabsorber (0.5, 1, 1.5, 2, 2.5, and 3%). Data are presented as the means ± standard deviations (*n* = 10; * *p* < 0.05, ** *p* < 0.01, and *** *p* < 0.001) and analyzed by a one-way ANOVA, followed by Tukey’s post hoc test. (**B**) Representative stress–strain curve of printed hydrogels with various concentrations of yellow photoabsorber (0.5, 1, 1.5, 2, 2.5, and 3%).

**Figure 5 ijms-23-05428-f005:**
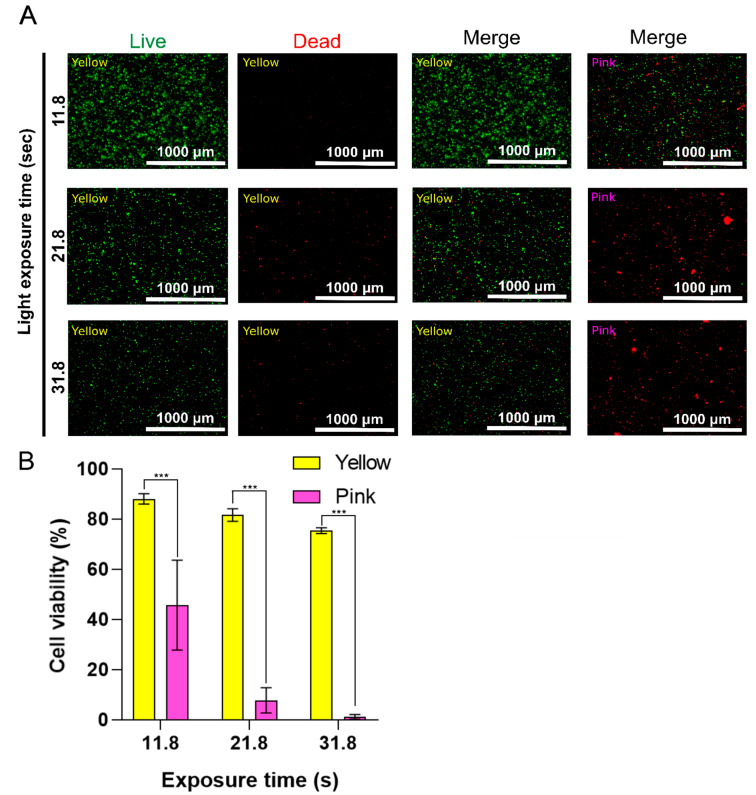
(**A**) Fluorescence microscopy images of a live/dead assay. BEFCs cultured in printed scaffolds after one day. Scale bar = 1000 µm. (**B**) Cell viability obtained from a live/dead assay of printed scaffold with yellow or pink pigment at different light-exposure time points: 11.8, 21.8, and 31.8 s. The quantification data are expressed as the means ± standard deviations (*n* = 5; *** *p* < 0.001). Statistical analyses were performed by using Student’s *t*-test.

**Figure 6 ijms-23-05428-f006:**
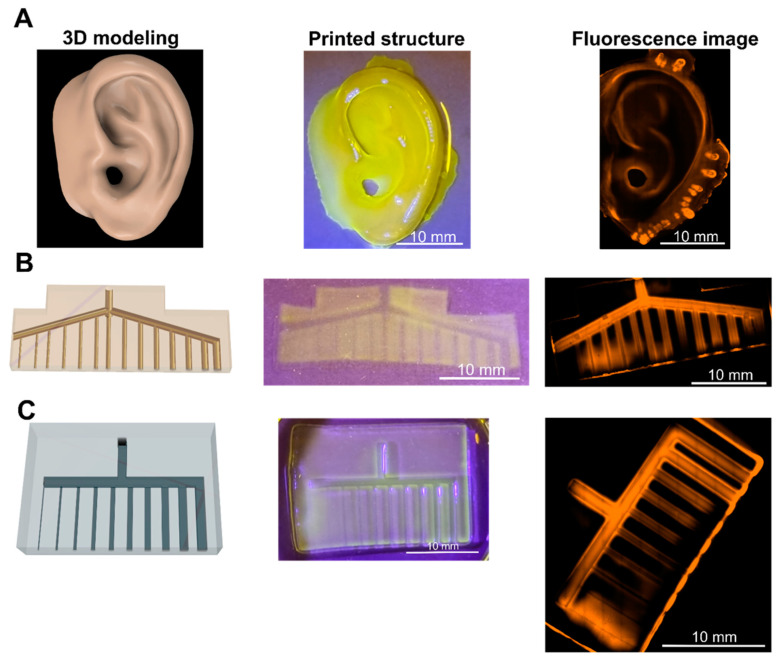
(**A**) Printed organ-like structure (ear). Scale bar = 10 mm. (**B**) Printed round-type microchannel structure (diameter: 100~1000 µm). Scale bar = 10 mm. (**C**) Printed square-type microchannel structure (diameter: 100~1000 µm). Scale bar = 10 mm.

## Data Availability

Data are contained within the article.

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
