# Peer review of "Improving Printability of Digital-Light-Processing 3D Bioprinting via Photoabsorber Pigment Adjustment"

_ijms, 2022, doi:10.3390/ijms23105428_

Round 1

Reviewer 1 Report

In their manuscript, Seo et al. aimed at optimizing the printing resolution by matching the absorption wavelength of a PEGDMA resin to the wavelength of their DLP printer. Even though the manuscript has an interesting point of view, its main findings are not new to the readers. As the LAP absorbs light at the UV and blue light range, it has always been clear based on the light absorption/emission theory that the yellow or orange color is needed to optimize the resolution of the LAP-based inks in blue light printers. Because the yellow color competes with the LAP initiator and reduces the penetration depth of the blue light, it is commonly used in the DLP printing. The manuscript contains some interesting data of the cell viability in differently colored resins. However, it unfortunately remains at a superficial level and does not try to understand the reasons behind the observed results. Because of the lack of proper discussion and understanding, the manuscript would work better as a brief communication instead of a full manuscript.

The manuscript raised several questions and comments that require authors’ attention:

  1. “Bioink” refers normally to printing materials that contain cells. To distinct an acellular ink from a cell-containing bioink, the better term could be “biomaterial ink”. (Please see Groll et al. A definition of bioinks and their distinction from biomaterial inks, Biofabrication 11, 2018)
  2. The description of the absorbance measurements is missing in the methods section. This definitely would be needed for better clarity.
  3. It was no clear, why the authors studied the effect of the color on a PEGDMA ink instead of the GelMA ink that was later used for the bioprinting. As the printing resolution strongly depends on the material itself, not only its color, GelMA should have been used for the resolution studies as well.
  4. In the rheometer results, it is not clear, what the authors mean with the sentence “At this point, the rheometer begins to sense less amount of liquid and thus low elasticity”. Why is the reduced contact to liquid decreasing the storage modulus and not increasing? Also, the second part is counterintuitive and requires more discussion: “As the viscosity increases, the loss modulus increases.” What is the theory behind this conclusion? Why does not the increasing viscosity increase the storage modulus?
  5. It was not explained, why the extended crosslinking time caused the cell death in the pink resin. What killed the cells, was it because of radicals or light energy?
  6. The statistical analysis was unclear. Does an ordinary one- and two-way analysis of variance mean ANOVA? In that case, how was Gaussian distribution confirmed? Where was the two-way analysis used?

Author Response

We have uploaded a Word file.

Reviewer 2 Report

The study "Improved Printability of Digital Light Processing 3D Bioprinting" describes a step-by-step optimization strategy for more accurate digital light processing (DLP) printing of biopolymers. The goal is to limit the crosslinking of prepolymers to a limited area and depth, while maintaining high cell viability. To reduce the trial and error of printing conditions, the authors proposed "synchronization" of the projection wavelength of the printer and the absorption wavelengths of the bioink using edible pigments as photoabsorbers. The authors showed how choosing the right pigment supports printing accuracy and cell viability. The optimal time for light exposure was rationally determined by photorheological measurements based on the gelling threshold. The study has great potential for publication in the International Journal of Molecular Sciences. The manuscript is well organized and written. However, there are several open questions that should be addressed by the authors for clarification.

  • What is the role and mechanism of the photoabsorbers?
    1. Is there any relevant energy transfer from absorbers to photoinitiator for the crosslinking process?
    2. Is it a coincidence that the spectra of the DLP projection and absorption of the yellow pigments overlap significantly?
    3. I am wondering whether this result indicates that the role of the photoabsorbers is just to reduce the intensity of the projected light. This might be an alternative explanation for the decreased over-cosslinking / scattering (which might result in a more accurate implementation rate) and increased cell viability (less harmful UV light reaches the cells). For clarification, control experiments are required to show whether or not similar success could be achieved by systematic reducing the intensity of the projection light.
  • Based on what criteria did the authors chose the concentration of pigments in section 2.1? The concentration might have an impact on the printing accuracy, too. Please comment.
  • The term "implementation rate" should be explained / defined in the results section. An alternative term could be pressure accuracy/ relative pressure deviation, etc.
  • Is the idea of a supplemented photoabsorber a new idea for DLP-based bioprinting? Please provide references.
  • What were the criteria to choose the pigments in this study? Please provide additional information on the chemical properties of the pigments.
  • The ability of DLP to print complex 3D structures was mentioned in the introduction. From my point of view, demonstrating the successful printing of a more complex structure would further strengthen the manuscript.

Author Response

We have uploaded a Word file.

Reviewer 3 Report

Overall, this manuscript investigate the effects of different photoabsorber color. Technically, it is useful. However, the writing is so weak, the title is not suitable.

  1. There are many methods to improve the printability, so just discussing the photoabsorber color, the title should be modifed.
  2. Some figures arrangements is so uncomfortable.
  3. DLP printer is not suitable, as DLP is the trademark,so projection base bioprinter or printing is more suitable. 
  4. Another discussion about the printablity of  projection base bioprinter should be cited, Printability during projection-based 3D bioprinting,Bioactive Materials,2022

Author Response

We have uploaded a Word file.

Round 2

Reviewer 1 Report

The revised manuscript is OK for publication.